# Nonadherence to anti-tuberculosis treatment, reasons and associated factors among pulmonary tuberculosis patients in the communities in Indonesia

**Dina Bisara Lolong**[1☯]*, **Ni Ketut Aryastami**[1☯], **Ina Kusrini**[1☯], **Kristina L. Tobing**[1☯], **Ingan Tarigan**[1☯], **Siti Isfandari**[1☯], **Felly Philipus Senewe**[1☯], **Raflizar**[1☯], **Noer Endah**[1☯], **Nikson Sitorus**[1☯], **Lamria Pangaribuan**[2‡], **Oster S. Simarmata**[2‡], **Yusniar Ariati**[2‡]

1 National Research and Innovation Agency (BRIN), Cibinong, Indonesia, 2 BKPK Indonesia Ministry of Health, Jakarta Selatan, Indonesia

☯ These authors contributed equally to this work.
‡ These authors contributed equally as co-authors.
* dina_lolong@yahoo.com

**Data Availability Statement:** All relevant data are within the paper and its Supporting Information files.

## Abstract

### Background

Tuberculosis (TB) is the world's major public health problem. We assessed the proportion, reasons, and associated factors for anti-TB treatment nonadherence in the communities in Indonesia.

### Methods

This national coverage cross-sectional survey was conducted from 2013 to 2014 with stratified multi-stage cluster sampling. Based on the region and rural-urban location. The 156 clusters were distributed in 136 districts/cities throughout 33 provinces, divided into three areas. An eligible population of age ≥15 was interviewed to find TB symptoms and screened with a thorax x-ray. Those whose filtered result detected positive followed an assessment of Sputum microscopy, LJ culture, and Xpert MTB/RIF. Census officers asked all participants about their history of TB and their treatment—defined Nonadherence as discontinuation of anti-tuberculosis treatment for <6 months. Data were analyzed using STATA 14.0 (College Station, TX, USA).

### Results

Nonadherence to anti-TB treatment proportion was 27.24%. Multivariate analysis identified behavioral factors significantly associated with anti-TB treatment nonadherence, such as smoking (OR = 1.78, 95% CI (1.47–2.16)); place of first treatment received: government hospital (OR = 1.45, 95% CI:1.06–1.99); private hospital (OR = 1.93, 95% CI: 1.38–2.72); private practitioner (OR = 2.24, 95% CI: 1.56–3.23); socio-demographic and TB status included region: Sumatera (OR = 1.44, 95% CI: 1.05–1.98); other areas (OR = 1.84, 95% CI: 1.30–2.61); low level of education (OR = 1.60, 95% CI: 1.27–2.03); and current TB positive status (OR = 2.17, 95% CI: 1.26–3.73).

**Funding:** This study was funded by Global Fund, KNCV, USAID and WHO. Each Funder had a specific role such as : Global Fund for data collection and management; KNCV and USAID for laboratory instruments including its apparatus; WHO for technical research assistant including training for the team, research preparation, data collection and supervision.

**Competing interests:** The authors have declared that no competing interests exist.

## Conclusions

Nonadherence to anti-TB drugs was highly related to the personal perception of the respondents, despite smoking, current TB status, a place for the first treatment, education, and region. The position of the first TB treatment at the private practitioner was significantly associated with the risk of Nonadherence to treatment.

## Introduction

Tuberculosis remains the leading cause of death from infectious disease among adults worldwide, with more than 10 million people becoming newly sick from tuberculosis yearly, while 1.5 million people die from TB. Indonesia is considered the third country with a high prevalence of TB after India and China [1] Free TB treatment following the directly observed treatment, short-course (DOTS) strategy is, among others, the primary process implemented in Indonesia [2]. Studies showed that DOTS increased the compliance rate, reduced the disease's recurrence, and prevented the development of multidrug resistance [3].

WHO launched the DOTS strategy in 1995; it comprises five significant components case detection by sputum smear microscopy, rapid molecular test, and standardized treatment with supervision and patient support [4]. Nonadherence to anti-TB treatment leads to increased length and severity of illness, death, disease transmission, and drug resistance, leading to a tremendous economic impact in terms of cost on patients as well as the health care system [5]. Nonadherence to TB treatment is one of the most significant obstacles to TB control globally and has become a major contributing factor to treatment failures [6]. A systematic review in developing countries mentioned socioeconomic demographic, lack of social support, and deficit of knowledge of the duration of treatment caused patients' loss of follow-up [7]. One smear-positive pulmonary TB patient is estimated to transmit the disease to an average of 10 other people per year [8]. Of these infected individuals, 10–12% will develop TB a few weeks or decades after infection [9]. Without treatment, about 70% of smear-positive and 20% of smear-negative patients will experience death over three years [10].

Treatment adherence of TB patients is a complex and multifaceted behavioral issue that needs to be understood better. However, few studies have examined Nonadherence to TB treatment, TB-related stigma, and its associated factors among TB patients in Indonesia. Therefore, we conducted this analysis to identify associated factors for Nonadherence to TB treatment among pulmonary tuberculosis patients in Indonesia based on the Indonesia 2013–2014 national TB prevalence survey.

## Materials and methods

### Study design and participants

This study used cross-sectional data from Indonesia's Tuberculosis Prevalence Survey from 2013 to 2014. The sampling method implemented a stratified multi-stage cluster sampling. The survey is conducted every ten years across the 33 provinces and 156 clusters by collaborating with the National Institute of Health Research and Development (NIHRD) and the Directorate General of Diseases Prevention and Control. A cluster comprises at least two census blocks of the population aged 15 years within approximately 500 people [11].

The sample was selected using multi-stage cluster sampling as follows:

- Stage 1: The probability proportional to size (PPS)-systematic sampling was implemented to select villages in each stratum. Census blocks of the geographic code sort the sampling frame of the villages.

- Stage 2: A cluster was selected randomly in each village. Exclusion criteria of the block census are institutional facilities such as military barracks and dormitories. This census involved 112,350 subjects from 156 population groups that met the requirements.

- Stage 3: Grouping 33 provinces in Indonesia into three regions, Java-Bali, Sumatra, others, and urban/rural classification. The eligible population included 76,576 subjects.

Samples were chosen from the eligible population. The inclusion criteria of the sample were those ages 15 years old and above, staying at least one month in the selected cluster. The exclusion criteria were those who were unwilling to participate in the survey. All participants signed informed consent before hands. For this analysis, we selected the sample diagnosed by a health worker as suffering from TB before and when the survey was not taking TB medication.

## Measurement of variable

The dependent variable was treatment adherence for participants who had a history of TB and experience of TB treatment for less than six months, while adherent TB was for participants who had a history of TB and experience of TB treatment for more than six months.

We categorized the independent variables with the operational definition as gender (male and female), age (15 to 54 and > 54 years old), and education (below high school and high school plus). Current TB status is the confirmed results of TB examination at the time of the survey (categorized as TB and non-TB cases). Contact history in the last two years with TB patient (yes or no); current smoking defined as still smoking at the time of the survey (yes or no).

The health center is the place of the first treatment received, categorized as the closest public health facility in the community. The second layer is the government hospital functional for patient referral, private hospital (other health facilities belong to the private sector), and private practitioners are defined as all clinics or health services by medic or paramedic including self-medication. The place of residents is categorized as urban and rural. Furthermore, the region is divided into Java-Bali, Sumatra, and others.

Reasons for Nonadherence refer to answers provided by respondents, such as feeling better, having no more symptoms or no improvement, recovering (declared cured by a health professional), fear of side effects, and other unacceptable reasons.

## Data analysis

Data were analyzed using STATA 14.0 (College Station, TX, USA). Associations between independent variables and the dependent variable of Nonadherence were estimated by calculating odds ratios (ORs) and their 95% confidence intervals (CIs) from the logistic regression model (Table 1). Predictive variables independently and significantly associated with treatment completion in univariate analysis were included in a multiple logistic regression model to determine their relative contributions in predicting treatment adherence while simultaneously adjusting for each effect. The criterion for significance was set at $P < 0.05$ based on a two-sided test.

## Ethics statement

The ethical clearance for the study was provided by the Ethics Committee of the NIHRD, number KE.01.10/EC/651/2012. This is secondary data analysis so that ethical clearance is no

**Table 1. Distribution and association between dependent and independent variables.**

| Variables | Total | Nonadherence n (%) | | OR (95% CI) | P value |
|---|---|---|---|---|---|
| | n | Yes | No | | |
| **Subject** | 2,045 | 557 (27.24%) | 1,488 (72.76%) | | |
| **Gender** | | | | | |
| Female | 905 | 214 (23.65) | 691 (76.35) | 1.00 | |
| Male | 1,140 | 343 (30.09) | 797 (69.91) | 1.41 (1.18–1.68) | 0.000 |
| **Age (years)** | | | | | |
| 15–54 | 1,490 | 380 (25.50) | 1,110 (74.50) | 1.00 | |
| ≥ 55 | 555 | 177 (31.89) | 378 (68.11) | 1.56 (1.05–1.67) | 0.018 |
| **Education** | | | | | |
| High | 658 | 147 (22.34) | 511 (77.66) | 1.00 | |
| Low | 1,387 | 410 (29.56) | 977 (70.44) | 1.43 (1.13–1.83) | 0.004 |
| **Current TB status** | | | | | |
| Non-TB | 1,985 | 529 (26.65) | 1,456 (73.35) | 1.00 | |
| Yes TB | 60 | 28 (46.67) | 32 (53.33) | 2.17 (1.25–3.74) | 0.006 |
| **History of contact with TB Patient** | | | | | |
| No | 1,823 | 508 (27.87) | 1,315 (72.13) | 1.00 | |
| Yes | 222 | 49 (22.07) | 173 (77.93) | 1.37 (0.99–1.89) | 0.054 |
| **Current smoking** | | | | | |
| No | 1,438 | 345 (23.99) | 1,093 (76.01) | 1.00 | |
| Yes | 607 | 212 (34.93) | 395 (65.07) | 1.72 (1.42–2.07) | 0.000 |
| **Place treatment was first received.** | | | | | |
| Health centres | 521 | 110 (21.11) | 411 (78.89) | 1.00 | |
| Government hospitals | 774 | 206 (26.61) | 568 (73.39) | 1.37 (1.02–1.89) | 0.039 |
| Private hospitals | 393 | 122 (31.04) | 271 (68.96) | 1.70 (1.21–2.39) | 0.002 |
| Private practitioners | 357 | 119 (33.33) | 238 (66.67) | 1.80 (1.26–2.58) | 0.001 |
| **Place of residence** | | | | | |
| Urban | 1,099 | 266 (24.20) | 833 (75.80) | 1.00 | |
| Rural | 946 | 291 (30.76) | 655 (69.24) | 1.40 (1.07–1.84) | 0.014 |
| **Region** | | | | | |
| Java-Bali | 967 | 224 (23.16) | 743 (76.84) | 1.00 | |
| Sumatera | 488 | 143 (29.30) | 345 (70.70) | 1.44 (1.06–1.97) | 0.022 |
| Other regions | 590 | 190 (32.20) | 400 (67.80) | 1.65 (1.17–2.31) | 0.004 |

more needed. Written informed consent was applied before the interview, and all participants signed it. All of the enumerators were trained to avoid bias in information. All identity details of the survey participants were kept confidential.

## Results

The total sample involved in the survey was 67,944 subjects. The number of issues included in this analysis was 2,191, with the criteria of those having a TB history. We further identified problems that had taken the TB treatment as 2,045 issues (see Fig 1).

We split the number into adherence and non-adherence to TB treatment, as seen in Fig 1.

We combined the univariate and bivariate analysis, as seen in Table 1. The results found that male participants in the older age group, who had a high level of education and lived in rural and outside Java-Bali, were more likely to be non-adherent. Participants currently diagnosed with TB mentioned no TB contact and were smoking were also more likely to be non-adherent. Comparing first places to seek treatment, we found that participants who went to

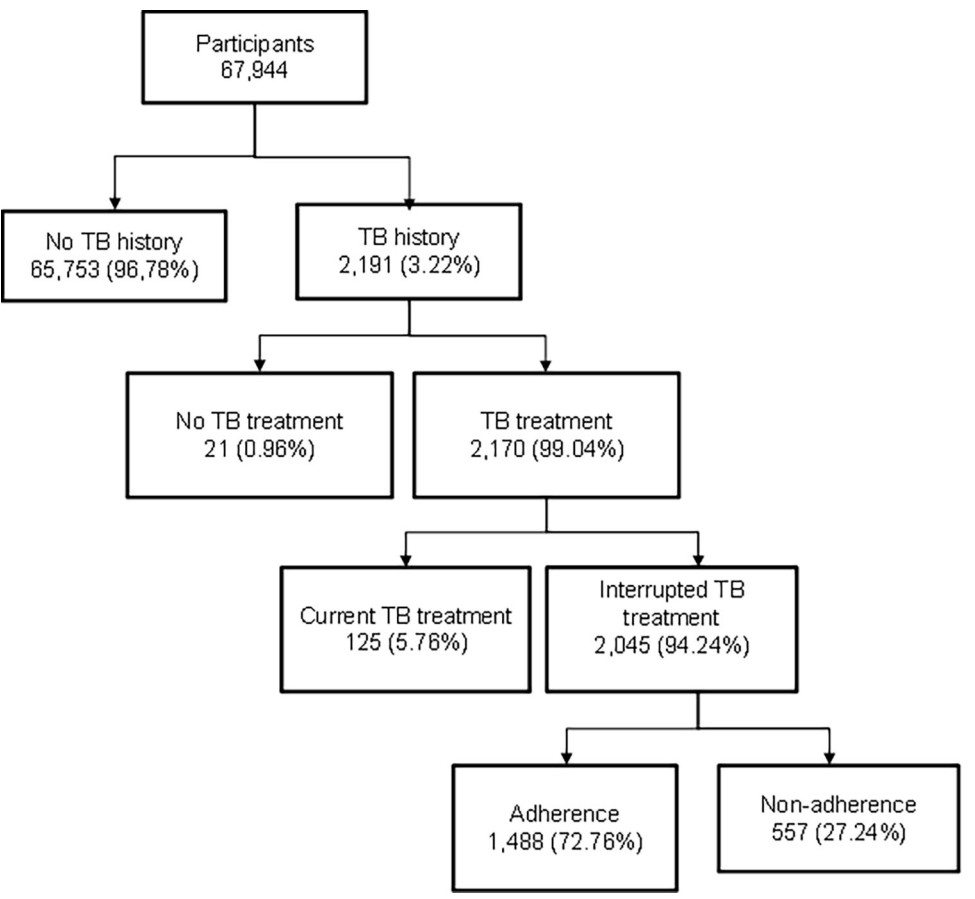

**Fig 1. Flowchart CONSORT diagram.** Note: The dotted lines are off concerned.

private practitioners had the highest OR of 1.80 (1.26–2.58), followed by private hospitals at 1.70 (1.21–2.39) and government hospitals at 1.37 (1.02–1.89), were more likely to be non-adherent compared to participants who went to health centers.

Table 1 shows the demographic variables of the male were more likely to Nonadherence to TB treatment, is 343 (30.09%) compared to females. i.e., 214 (23.65%). This result was significant by the odds ratio of OR = 1.41 (95% CI: 1.18–1.68). The age of the respondent also showed that older age ($\geq$ 55 years) was more likely to 177 (31.89%) nonadherences than the younger (15–45 years) 378 (68.11%) OR = 1.56 (95% CI: 1.05–1.67). In terms of education, those with lower education were significantly more likely to have non-adherence to TB treatment.

The proportion of positive cases in this survey was 2.93%. Nearly half (46.67%) of the TB-positive respondents significantly Nonadherence to TB treatment (OR = 2.7; 95% CI: 1.25–3.74). The contact history of the respondent with an ever-TB patient is substantially more likely adherence to TB treatment (OR = 1.37; 95% CI: 0.99–1.89).

Government Hospital was the most utilized place to seek TB treatment (37.85%), followed by Health Center (25.48%). The subject which, for the first time, got treatment from private practitioners mostly had Nonadherence to TB treatment respectively (OR = 1.80; 95% CI: 1.26–2.58). Regarding smoking, as much as 70.32% of the subject was not currently smoking. Those smoking (34.93%) were significantly Nonadherence compared to the adherence one (OR = 1.72; 95% CI: 1.42–2.07).

**Table 2. Multivariate logistic regression analysis of nonadherence to TB treatment.**

| Variables | OR (95% CI) | P-value |
|---|---|---|
| **Education** | | |
| High | 1.00 | |
| Low | 1.60 (1.27–2.03) | 0.000 |
| **Current TB status** | | |
| Non-TB | 1.00 | |
| Yes TB | 2.17 (1.26–3.73) | 0.005 |
| **Current smoking** | | |
| No | 1.00 | |
| Yes | 1.78 (1.47–2.16) | 0.000 |
| **Place treatment was first received.** | | |
| Health centers | 1.00 | |
| Government hospitals | 1.45 (1.06–1.99) | 0.021 |
| Private hospitals | 1.93 (1.38–2.72) | 0.000 |
| Private practitioners | 2.24 (1.56–3.23) | 0.000 |
| **Region** | | |
| Java-Bali | 1.00 | |
| Sumatera | 1.44 (1.05–1.98) | 0.025 |
| Other regions | 1.84 (1.30–2.61) | 0.001 |

The proportion of subjects living in urban was higher (53.74%) than in rural areas. Issues that live in rural areas (30.76%) compared to urban areas (24.20%) are significantly almost one and a half fold more likely to be Nonadherence (OR = 1.40; 95% CI: 1.07–1.84). The region has a significant association with Nonadherence to TB treatment. The area of outer Jawa-Bali has a higher risk of Nonadherence, i.e., Sumatera OR = 1.44 (95% CI: 1.06–1.97), and other regions OR = 1.65 (95% CI: 1.17–2.31).

The results of multivariate logistic regression analysis depicted in Table 2 found that the level of education, current TB status, current smoking, the first place to seek treatment, and region were significantly associated factors after controlling each other. Participants with a lower level of education were 1.6 times (OR = 1.6, 95% CI: 1.27–2.03) more likely to be Nonadherence to their anti-TB therapy than those with higher educational levels. Participants who were positive for TB at the time of the survey were 2.17 times (OR = 2.17; 95% CI: 1.26–3.73) more likely to be Nonadherence to TB treatment than participants who were non-TB treatment at the time of the survey. Participants who were Nonadherence to TB treatment were 2.17 times (OR = 2.17; 95% CI: 1.26–3.73) more current TB positive. Participants with current smoking were 1.78 times (OR = 1.78, 95% CI: 1.47–2.16) more likely to be Nonadherence to their anti-TB therapy than non-smoking participants. Furthermore, participants who went to seek treatment at government hospitals, private hospitals, and private practitioners were 1.45 times (OR = 1.45, 95% CI: 1.06–1.99), 1.93 times (OR = 1.93, 95% CI: 1.38–2.72), and 2.24 times (OR = 2.24, 95% CI: 1.56–3.23), respectively more likely to be Nonadherence to their anti-TB treatment compared to participants who went to health centers.

In addition, participants who lived in Sumatera and other regions were 1.44 times (OR = 1.44, 95% CI: 1.05–1.98) and 1.84 times (OR = 1.84, 95% CI: 1.30–2.61), respectively more likely to be Nonadherence to their anti-TB therapy than participants who lived in Java-Bali.

Participants were asked about reasons for the duration of TB medications before six months. The three main reasons for Nonadherence listed by patients, as shown in Table 3,

**Table 3. The main reason for Nonadherence listed by participants.**

|     | Reasons for Nonadherence | n* | % |
|-----|--------------------------|-----|------|
| 1.  | Feeling better/ there were no more symptoms | 223 | 40.04 |
| 2.  | Declared cured by a health professional | 142 | 25.49 |
| 3.  | Having no money | 71 | 12.75 |
| 4.  | Fear of side effects | 29 | 5.21 |
| 5.  | Feeling no improvement | 27 | 4.85 |
| 6.  | Nobody took the medicine | 22 | 3.95 |
| 7.  | No transportation means | 15 | 2.69 |
| 8.  | No TB drugs are available in the health facility | 7 | 1.26 |
| 9.  | Other reasons | 21 | 3.77 |

*: Total number of non-adherent participants = 557.

were feeling better/there were no more symptoms 223 (40.04%), declared cured by a health professional 142 (25.49%), and having no money 71(12.75%). The other reasons were fear of the TB drugs' side effects 29 (5.21%), not feeling any improvement 27 (4.85%), and other reasons, as seen in Table 3.

## Discussion

In this study, the rate of Nonadherence to anti-tuberculosis treatment was 27.2%. This aligns with the study done at Gondar town health centers in Northwest Ethiopia [12] and at TB clinics in Arba Minch Government Health Institutions, Southern Ethiopia [13], which reported 21.2% and 24.7% rates, respectively. However, it is higher than in studies done in the community in PR China [14] and Bandung at the TB lung clinic, Indonesia [15], which were 12.2% and 16%, respectively. This finding is lower than in studies conducted at TB treatment centers in India (50%) [16] and Mekelle, Ethiopia (55.8%) [17]. The variation might be due to the differences in study design and settings.

It is well known from reality that treatment results in feeling better or symptoms relieved after a few weeks of receiving it. We found in this study that the most frequently mentioned reason for Nonadherence to treatment was feeling better or cured. Similar findings were reported from Indonesia [18,19] and other countries [12,20,21].

This study showed that low education was associated with Nonadherence to TB treatment (OR = 1.6; 95% CI: 1.27–2.03). This is different from other studies, which show that the level of education is not significantly associated with adherence [22]. Our study found that participants who received TB treatment for the first time at private practitioners have a 2.24 times greater risk of Nonadherence than those who received treatment for the first time at a Health Center. Surprisingly, one of the reasons for not being compliant with TB treatment was being declared cured by a health professional, which accounted for almost a quarter of the participants. This study agrees that Nonadherence to TB treatment was the highest among patients who went to private practitioners for the first treatment. In Indonesia, most private practitioners (97%) still need to implement the DOTS strategy emphasizing TB treatment for at least six months [23]. Besides no transportation means and nobody collecting medicine, this study revealed that the reason for stopping TB treatment was financial problems, even though the drugs were free. Challenges in accessing healthcare due to reasons such as difficult areas or the cost of seeking care have been reported by several studies, including in Indonesia, as risk factors for Nonadherence to TB treatment [15,18,19,24,25].

This study's findings showed that smoking was one of the individual behavioral factors significantly associated with TB treatment nonadherence (OR = 1.78; 95% CI: 1.45–2.16). This may occur because smokers think that chronic cough is only due to smoking behavior. This finding was consistent with studies conducted in other developing countries [26,27]. A study in Mumbai, India, found that participants who smoked had a 2.4 times risk of not adhering to treatment than non-smokers [27]. Nonadherence to TB treatment was also associated with living in rural areas and other regions (outside Java-Bali and Sumatera). Other parts, such as Papua Province in Indonesia, mostly need more access to healthcare due to geographical constraints.

Furthermore, participants who received their first treatment at the primary health care (PHC) closest to their home were less likely to be non-adherent to TB treatment than the ones who received their first treatment at hospitals and private practitioners. Ruru et al. found that the main factor associated with Nonadherence during tuberculosis treatment among TB patients in Jayapura, Papua province, was the problem of distance [24]. In Nigeria, the rural residence was a predictor of treatment default in a tertiary hospital, undoubtedly due to the distance from home to urban clinics [28]. Similarly, a study in Arba Minch Government Health Institutions, Southern Ethiopia, found that the highest factor associated with Nonadherence to TB treatment was the residence that was too far from the health facility [13].

One identified independent risk factor for Nonadherence was the male gender, which was similar to other studies [16,18,29]. The male is the leading economic provider for their Indonesian family. Thus, they tend to leave home early for work and may have difficulty complying with treatment and follow-up [12,30].

This study showed that participants suffering from the results of sputum examinations were 2.17 times less compliant with TB treatment compared to participants who did not suffer from it at the time of the survey (OR = 2.17; 95% CI: 1.26–3.73). Nonadherence to TB treatment was significantly associated with recurrent or relapse tuberculosis patients. A high relapse rate may indicate unsuccessful treatment [31]. A study in China, which may be similar to Indonesia, demonstrated more relapse cases than reinfection cases [32]. Poor adherence contributes to the worsening TB situation by increasing drug resistance. A recent study in Indonesia in 2018 demonstrated that retreated TB patients were around nine times higher than new TB patients (12.4% VS 1.4%) [33].

The age variable was another factor associated with TB treatment nonadherence. Similar to our study, some studies, such as in Nepal [34], found that older age was significantly related to the risk of Nonadherence to TB treatment. Conversely, a study in Jayapura, Indonesia [24] found that the younger generation was associated with Nonadherence, while a survey in Bandung [15], Indonesia, did not see any difference between age groups. Older people might assume more family responsibility, or due to their weakness, they may require family support or another reason was a geographic problem that tends to stop medications.

Adverse effects were also found to be the reason for Nonadherence in this study, in concurrence with other studies [35–38]. Therefore, patients should be informed about the side effects to maintain compliance. These findings showed that the proportion of Nonadherence was a challenge to ending the TB epidemic targeted by SDGs and the End TB strategy by 2030 and 2035, respectively [39].

## Conclusions

This study found that the proportion of Nonadherence to anti-TB drugs was high in the communities in Indonesia. The place for first TB treatment at the private practitioner is significantly associated with Nonadherence to treatment. Reasons for Nonadherence to TB

treatment depend on subject response and self-confession, respectively.The community is undisciplined in TB treatment. Thus, the role of health workers is essential to focus more on monitoring and implementing standard therapy according to SOP.

## Limitations

The design of this study is cross-sectional, so the temporal relationship of several independent variables with the dependent variable cannot be ascertained. Nonadherence was assessed from the TB treatment history instead of hospital-based data. Consequently, participants might be subjected to recall bias. Reasons for Nonadherence to TB treatment are only based on subject response and self-confession as one of the study limitations.

## Supporting information

**S1 File.**
(XLSX)

**S2 File.**
(PDF)

## Author Contributions

**Conceptualization:** Dina Bisara Lolong, Noer Endah.

**Data curation:** Dina Bisara Lolong, Ni Ketut Aryastami, Kristina L. Tobing, Nikson Sitorus, Lamria Pangaribuan, Oster S. Simarmata.

**Formal analysis:** Dina Bisara Lolong, Ni Ketut Aryastami, Kristina L. Tobing, Ingan Tarigan, Siti Isfandari, Noer Endah, Nikson Sitorus, Lamria Pangaribuan, Oster S. Simarmata.

**Investigation:** Raflizar, Noer Endah, Yusniar Ariati.

**Methodology:** Dina Bisara Lolong, Ni Ketut Aryastami, Kristina L. Tobing, Siti Isfandari, Felly Philipus Senewe, Raflizar, Lamria Pangaribuan, Oster S. Simarmata.

**Supervision:** Felly Philipus Senewe.

**Validation:** Ina Kusrini, Ingan Tarigan, Yusniar Ariati.

**Writing – original draft:** Dina Bisara Lolong.

**Writing – review & editing:** Dina Bisara Lolong, Ni Ketut Aryastami, Ina Kusrini, Kristina L. Tobing, Ingan Tarigan, Siti Isfandari, Felly Philipus Senewe, Oster S. Simarmata, Yusniar Ariati.

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
