## [Decision Letter · Decision Letter 0]

8 Feb 2022

PONE-D-21-23837Non-Adherence to Anti-Tuberculosis Treatment, Reasons and Associated Factors among Pulmonary Tuberculosis Patients in The Communities in IndonesiaPLOS ONE

Dear Dr. Kusrini,

Thank you for submitting your manuscript to PLOS ONE. After careful consideration, we feel that it has merit but does not fully meet PLOS ONE’s publication criteria as it currently stands. Therefore, we invite you to submit a revised version of the manuscript that addresses the points raised during the review process. Please pay a good read on the review. Major flaws have been detected specially in material and methods. Please submit your revised manuscript by Mar 24 2022 11:59PM. If you will need more time than this to complete your revisions, please reply to this message or contact the journal office at plosone@plos.org. Please include the following items when submitting your revised manuscript:A rebuttal letter that responds to each point raised by the academic editor and reviewer(s). You should upload this letter as a separate file labeled 'Response to Reviewers'.A marked-up copy of your manuscript that highlights changes made to the original version. You should upload this as a separate file labeled 'Revised Manuscript with Track Changes'.An unmarked version of your revised paper without tracked changes. You should upload this as a separate file labeled 'Manuscript'.

We look forward to receiving your revised manuscript.

Kind regards,

Pere-Joan Cardona, MD, PhD

Academic Editor

PLOS ONE

Journal Requirements:

2. Please include a copy of the interview guide used in the study, in both the original language and English, as Supporting Information, or include a citation if it has been published previously.

3. Please provide additional details regarding participant consent. In the ethics statement in the Methods and online submission information, please ensure that you have specified:

 - whether consent was obtained

 - whether consent was informed 

 - what type of consent you obtained (for instance, written or verbal, and if verbal, how it was documented and witnessed). 

 - if your study included minors, state whether you obtained consent from parents or guardians. 

 - if the need for consent was waived by the ethics committee, please include this information.

NO, The funders had no role in study design, data collection and analysisi, decision to publish, or preparation of the manuscript

Reviewers' comments:

Reviewer's Responses to Questions

**Comments to the Author**

1. Is the manuscript technically sound, and do the data support the conclusions?

Reviewer #1: Partly

2. Has the statistical analysis been performed appropriately and rigorously? 

Reviewer #1: I Don't Know

3. Have the authors made all data underlying the findings in their manuscript fully available?

Reviewer #1: Yes

4. Is the manuscript presented in an intelligible fashion and written in standard English?

Reviewer #1: Yes

5. Review Comments to the Author

Reviewer #1: The text is very interesting but, in my opinion, is necessary to clarify some aspects about it, and review the parts for give more cohesion to the presented work.

As a few general recommendations, please, see below.

Please, review the text and definitions.

According to the principal aim of the study, the authors should review the conclusions of the work. According to the title, reference is made to reasons and factors associated with non-adherence to tuberculosis treatment. However, the conclusions do not meet this expectation.

In the introduction, reference is made to the DOTs program; however, the procedure for controlling the treatment is not explained. And I think it is important in a work on adherence to treatment. This is not explained in the methods section either.

En materials and Methods I have some questions.

Please, could the authors clarify what is the sense of "stratified multi-stage clusters sampling"? What were the inclusion criteria for introducing the patients in the study? What was the significance of "selected cluster"?

How the participants in the study were selected? Were they enrolled consecutively?

What was the usual duration of treatments and the used drugs?

What was the timing of treatment regarding to the interview and inclusion in the Study?

What was the criterion for non-adherence?

It would be helpful to define the terms that will be evaluated during the Study. Please, add.

Were confirmed cases from possible cases considered separately in the analysis?

Please, clarify the criteria for non-adherence. Likewise, establish clear criteria on adherence to treatment and non-compliance. Also, the criteria to establish the categories of cured, treatment completed, lost, treatment failure (regardless of adherence to treatment)

In data analysis part, the definition criteria should be moved to the previous part.

In results, the first part is about proportion of non-adherence. However, it could be better to explain the most important findings of the study or, in any case, the number of cases included. Moreover, the authors used the expression of proportion when they talk about total number. Please, review.

Any enrolled patient was excluded of the analysis?

The second part of results is called “factors associated with-non-adherence to anti-TB therapy”. It seems that the objective and subjective data were not analyzed separately.

It seems that, patients in current TB are included in the study. It is true? In this case, please, would you clarify the term: "Non TB cases vs "Yes TB cases".

Regarding the “Current smoking”, it was a condition during the TB treatment or at the moment of completing the interview?

Please, explain the differences about “health centers”, Government hospitals”, “private hospitals”, Private practitioners”.

Regarding the interview, was the interview performed by patients or personnel?

In relation to interrupted treatment, on the first line, the authors, talk about “interruption of TB medications before six months”, please, clarify.

On the discussion part, start with the more significant findings. It seems there are three elements: the proportion of non-adherence, the associated general factors and the “reasons for discontinuing treatment. Please, clarify every “part”. In my opinion it would be clearer to avoid mixture of data.

Please, listening the associated factors or reasons related to non-adherence in the beginning of every part would be easier to know the findings of the Study.

Was the survey data checked (for example, when they answered that the professional told them they were cured)?

On the conclusions, please, consider the title of the paper. The Study and the conclusions should respond the principal aim and expectative created by the title of the paper. Non-adherence factors are mentioned and the actions to reduce the interruption of treatment were introduced for first time. Please, think to talk about it in the discussion part, for instance and in the introduction.

“Ethics statement” should be moved before the results, due to, in my opinion, is related to Methods.

Please, review the bibliography and the number of authors per reference.

6. PLOS authors have the option to publish the peer review history of their article (what does this mean?). If published, this will include your full peer review and any attached files.

Reviewer #1: **Yes: **Antonio Moreno

---

## [Author Response · Author response to Decision Letter 0]

26 Jul 2022

We appreciate your suggestion. We appreciate all of the feedback from reviewers and an academic editor. To address the query or all requirements, we have included a response letter to the editor and reviewer in this file. Respond to editor in two files as other files. Respond to editor 1 and respond to editor 2

---

## [Decision Letter · Decision Letter 1]

27 Sep 2022

PONE-D-21-23837R1Non-Adherence to Anti-Tuberculosis Treatment, Reasons and Associated Factors among Pulmonary Tuberculosis Patients in The Communities in IndonesiaPLOS ONE

Dear Dr. Kusrini,

Thank you for submitting your manuscript to PLOS ONE. After careful consideration, we feel that it has merit but does not fully meet PLOS ONE’s publication criteria as it currently stands. Therefore, we invite you to submit a revised version of the manuscript that addresses the points raised during the review process. Please follow the comments of the reviewer regarding on conclusions and references.

We look forward to receiving your revised manuscript.

Kind regards,

Pere-Joan Cardona, MD, PhD

Academic Editor

PLOS ONE

Journal Requirements:

Reviewers' comments:

Reviewer's Responses to Questions

**Comments to the Author**

1. If the authors have adequately addressed your comments raised in a previous round of review and you feel that this manuscript is now acceptable for publication, you may indicate that here to bypass the “Comments to the Author” section, enter your conflict of interest statement in the “Confidential to Editor” section, and submit your "Accept" recommendation.

Reviewer #1: (No Response)

2. Is the manuscript technically sound, and do the data support the conclusions?

Reviewer #1: Yes

3. Has the statistical analysis been performed appropriately and rigorously? 

Reviewer #1: I Don't Know

4. Have the authors made all data underlying the findings in their manuscript fully available?

Reviewer #1: Yes

5. Is the manuscript presented in an intelligible fashion and written in standard English?

Reviewer #1: Yes

6. Review Comments to the Author

Reviewer #1: Thank you for the explanations and corrections on the paper.

I don't have much more questions regarding it.

Only a few details: On the part "Study design and participants", the sentence "written informed consent was obtained and it was signed to all participants" could be written without bold letters. On the part "Ethics statement" the informed consent process is explained again.

Finally, on "conclusions", at the end of first paragraph the authors say: "Reasons for non-adherence TB treatment depend on subject response and self-confession respectively". According to the methods it could be considered as limitation of the study, and it is reported in this sense on the part "limitations".

On the other hand, on the part of “References”:

#1, please, the link to the document should be added.

#2, please, the link to the document should be added.

#4, please, the link to the document should be added.

Please, review the reference #5. The authors are: Dick, J.; Lombard, C.

Please, the link of #7 should be added.

Please, the link of #11 should be added.

Please, the link of #23 should be added.

Please, review if it is necessary to add new links on the reference part, as, for instance, on #40

7. PLOS authors have the option to publish the peer review history of their article (what does this mean?). If published, this will include your full peer review and any attached files.

Reviewer #1: No

---

## [Author Response · Author response to Decision Letter 1]

20 Oct 2022

Thank you for your review. We have revised our manuscript follow your suggestion. We attached the clean manucsript, manuscript with track change and rebuttal letter for reviewer 20 Oct 2022

---

## [Editor Report · Decision Letter 2]

17 Nov 2022

PONE-D-21-23837R2Non-Adherence to Anti-Tuberculosis Treatment, Reasons and Associated Factors among Pulmonary Tuberculosis Patients in The Communities in IndonesiaPLOS ONE

Dear Dr. Kusrini,

Thank you for submitting your manuscript to PLOS ONE. After careful consideration, we feel that it has merit but does not fully meet PLOS ONE’s publication criteria as it currently stands. Therefore, we invite you to submit a revised version of the manuscript that addresses the points raised during the review process. Please provide the links to the references and fix the grammar issues.

We look forward to receiving your revised manuscript.

Kind regards,

Pere-Joan Cardona, MD, PhD

Academic Editor

PLOS ONE
---

## [Author Response · Author response to Decision Letter 2]

5 Jan 2023

we have revised following the requirement, we added response for reviewer for rebuttal letter in attacment file and we upload clean manuscript with some changed in reference. We have stated all contribution of funder in cover letter and respond to editor

---

## [Decision Letter · Decision Letter 3]

15 Mar 2023

PONE-D-21-23837R3Non-Adherence to Anti-Tuberculosis Treatment, Reasons and Associated Factors among Pulmonary Tuberculosis Patients in The Communities in IndonesiaPLOS ONE

Dear Dr. Kusrini,

Thank you for submitting your manuscript to PLOS ONE. After careful consideration, we feel that it has merit but does not fully meet PLOS ONE’s publication criteria as it currently stands. Therefore, we invite you to submit a revised version of the manuscript that addresses the points raised during the review process.

Please submit your revised manuscript by Apr 29 2023 11:59PM. If you will need significantly more time than this to complete your revisions, please reply to this message or contact the journal office at plosone@plos.org. Please include the following items when submitting your revised manuscript:A rebuttal letter that responds to each point raised by the academic editor and reviewer(s). You should upload this letter as a separate file labeled 'Response to Reviewers'.A marked-up copy of your manuscript that highlights changes made to the original version. You should upload this as a separate file labeled 'Revised Manuscript with Track Changes'.An unmarked version of your revised paper without tracked changes. You should upload this as a separate file labeled 'Manuscript'.If applicable, we recommend that you deposit your laboratory protocols in protocols.io to enhance the reproducibility of your results. Protocols.io assigns your protocol its own identifier (DOI) so that it can be cited independently in the future. For instructions see: https://journals.plos.org/plosone/s/submission-guidelines#loc-laboratory-protocols. Additionally, PLOS ONE offers an option for publishing peer-reviewed Lab Protocol articles, which describe protocols hosted on protocols.io. Read more information on sharing protocols at https://plos.org/protocols?utm_medium=editorial-email&utm_source=authorletters&utm_campaign=protocols.

We look forward to receiving your revised manuscript.

Kind regards,

Frederick Quinn

Academic Editor

PLOS ONE

Journal Requirements:

Reviewers' comments:

Reviewer's Responses to Questions

**Comments to the Author**

1. If the authors have adequately addressed your comments raised in a previous round of review and you feel that this manuscript is now acceptable for publication, you may indicate that here to bypass the “Comments to the Author” section, enter your conflict of interest statement in the “Confidential to Editor” section, and submit your "Accept" recommendation.

Reviewer #1: (No Response)

2. Is the manuscript technically sound, and do the data support the conclusions?

Reviewer #1: Yes

3. Has the statistical analysis been performed appropriately and rigorously? 

Reviewer #1: I Don't Know

4. Have the authors made all data underlying the findings in their manuscript fully available?

Reviewer #1: Yes

5. Is the manuscript presented in an intelligible fashion and written in standard English?

Reviewer #1: Yes

6. Review Comments to the Author

Reviewer #1: The text was improved; however, I think it needs a few minimal changes.

In some parts there is repetitions and data that could be expressed avoiding redundancies.

For instance, please, review the introduction section and avoid terms or repetitive concepts. This action, will improve the comprehension and the ease of reading.

Similar aspects should be reviewed on the part “Study design and participants”.

Age, studies carried out, are repeated later. Please, review.

On the last part, the paragraph: "the total sample involved in the survey..." should be moved to the part of results. In fact, now it is started doing reference to this data.

Please, review the part of “measurement of variable” and avoid the duplicity of data.

I have a few questions regarding the number of included patients.

Were all patients included in the study? Were no patients lost during the study?

No patients were excluded?

Consider removing some of the non-essential data in the text, where the data is already in the tables. It makes reading more difficult and does not provide more information. respect to work.

Regarding the changes performed in the text due to the revision process, some verbs and sentences should be changed now

Please, clarify the second conclusion of this part.

7. PLOS authors have the option to publish the peer review history of their article (what does this mean?). If published, this will include your full peer review and any attached files.

Reviewer #1: **Yes: **Antonio Moreno

---

## [Author Response · Author response to Decision Letter 3]

30 Apr 2023

Thank you for a valuable review. We have revised following the reviewer comments, and we attach clean and trackchange manuscript also rebuttal letter. Thank you. Hopefully, we have good news of our submitted manuscript.

---

## [Decision Letter · Decision Letter 4]

13 Jun 2023

Non-Adherence to Anti-Tuberculosis Treatment, Reasons and Associated Factors among Pulmonary Tuberculosis Patients in The Communities in Indonesia

PONE-D-21-23837R4

Dear Dr. Kusrini,

We’re pleased to inform you that your manuscript has been judged scientifically suitable for publication and will be formally accepted for publication once it meets all outstanding technical requirements.

Kind regards,

Frederick Quinn

Academic Editor

PLOS ONE

Additional Editor Comments (optional):

Reviewers' comments:

Reviewer's Responses to Questions

**Comments to the Author**

1. If the authors have adequately addressed your comments raised in a previous round of review and you feel that this manuscript is now acceptable for publication, you may indicate that here to bypass the “Comments to the Author” section, enter your conflict of interest statement in the “Confidential to Editor” section, and submit your "Accept" recommendation.

Reviewer #1: All comments have been addressed

2. Is the manuscript technically sound, and do the data support the conclusions?

Reviewer #1: Yes

3. Has the statistical analysis been performed appropriately and rigorously? 

Reviewer #1: I Don't Know

4. Have the authors made all data underlying the findings in their manuscript fully available?

Reviewer #1: Yes

5. Is the manuscript presented in an intelligible fashion and written in standard English?

Reviewer #1: Yes

6. Review Comments to the Author

Reviewer #1: No tengo comentarios adicionales a hacer. Ahora, el artículo es mucho más sencillo de leer y se ajusta mejor a lo establecido en el título y objetivos del mismo. Los puntos fundamentales del trabajo se muestran con mayor claridad al lector.

7. PLOS authors have the option to publish the peer review history of their article (what does this mean?). If published, this will include your full peer review and any attached files.

Reviewer #1: No

---

## [Editor Report · Acceptance letter]

27 Jul 2023

PONE-D-21-23837R4 

Nonadherence to Anti-Tuberculosis Treatment, Reasons and Associated Factors among Pulmonary Tuberculosis Patients in The Communities in Indonesia 

Dear Dr. Kusrini:

I'm pleased to inform you that your manuscript has been deemed suitable for publication in PLOS ONE. Congratulations! Your manuscript is now with our production department. 

Kind regards, 

on behalf of

Dr. Frederick Quinn 

Academic Editor

PLOS ONE